# Surveillance and Characterization of Vancomycin-Resistant and Vancomycin-Variable Enterococci in a Hospital Setting

**DOI:** 10.3390/antibiotics14080795

**Published:** 2025-08-04

**Authors:** Claudia Rotondo, Valentina Antonelli, Alberto Rossi, Silvia D’Arezzo, Marina Selleri, Michele Properzi, Silvia Turco, Giovanni Chillemi, Valentina Dimartino, Carolina Venditti, Sara Guerci, Paola Gallì, Carla Nisii, Alessia Arcangeli, Emanuela Caraffa, Stefania Cicalini, Carla Fontana

**Affiliations:** 1Microbiology and Biobank Unit, National Institute for Infectious Diseases “Lazzaro Spallanzani”, IRCCS, 00149 Rome, Italy; claudia.rotondo@inmi.it (C.R.); valentina.antonelli@inmi.it (V.A.); alberto.rossi@inmi.it (A.R.); silvia.darezzo@inmi.it (S.D.); marina.selleri@inmi.it (M.S.); michele.properzi@inmi.it (M.P.); valentina.dimartino@inmi.it (V.D.); carolina.venditti@inmi.it (C.V.); saraguerci98@gmail.it (S.G.); carla.fontana@inmi.it (C.F.); 2Department of Agriculture and Forest Sciences (DAFNE), University of Tuscia, 01100 Viterbo, Italy; silvia.turco@unitus.it; 3Bioinformatics Research Unit in Infectious Diseases, National Institute for Infectious Diseases “Lazzaro Spallanzani”, IRCCS, 00149 Rome, Italy; giovanni.chillemi@inmi.it; 4Department of Experimental Medicine, University of Rome “Tor Vergata”, 00133 Rome, Italy; 5Clinical Risk Management, National Institute for Infectious Diseases “Lazzaro Spallanzani”, IRCCS, 00149 Rome, Italy; paola.galli@inmi.it (P.G.); alessia.arcangeli@inmi.it (A.A.); 6Systemic and Immune Depression-Associated Infections Unit, National Institute for Infectious Diseases “Lazzaro Spallanzani”, IRCCS, 00149 Rome, Italy; emanuela.caraffa@inmi.it (E.C.); stefania.cicalini@inmi.it (S.C.)

**Keywords:** enterococci, vancomycin-resistant, vancomycin-variable, whole-genome sequencing, antimicrobial resistance, hospital surveillance

## Abstract

**Background/Objectives**: Enterococci, particularly *Enterococcus faecalis* and *Enterococcus faecium*, are Gram-positive cocci that can cause severe infections in hospitalized patients. The rise of vancomycin-resistant enterococci (VRE) and vancomycin-variable enterococci (VVE) poses significant challenges in healthcare settings due to their resistance to multiple antibiotics. **Methods**: We conducted a point prevalence survey (PPS) to assess the prevalence of VRE and VVE colonization in hospitalized patients. Rectal swabs were collected from 160 patients and analyzed using molecular assays (MAs) and culture. Whole-genome sequencing (WGS) and core-genome multilocus sequence typing (cgMLST) were performed to identify the genetic diversity. **Results**: Of the 160 rectal swabs collected, 54 (33.7%) tested positive for the *van*A and/or *van*B genes. Culture-based methods identified 47 positive samples (29.3%); of these, 44 isolates were identified as *E. faecium* and 3 as *E. faecalis*. Based on the resistance profiles, 35 isolates (74.5%) were classified as VRE, while 12 (25.5%) were classified as VVE. WGS and cgMLST analyses identified seven clusters of *E. faecium*, with sequence type (ST) 80 being the most prevalent. Various resistance genes and virulence factors were identified, and this study also highlighted intra- and inter-ward transmission of VRE strains. **Conclusions**: Our findings underscore the potential for virulence and resistance of both the VRE and VVE strains, and they highlight the importance of effective infection control measures to prevent their spread. VVE in particular should be carefully monitored as they often escape detection. Integrating molecular data with clinical information will hopefully enhance our ability to predict and prevent future VRE infections.

## 1. Introduction

Enterococci, such as *Enterococcus faecalis* and *Enterococcus faecium*, are Gram-positive, facultatively anaerobic cocci typically found in the gastrointestinal tracts of humans and animals. While they are a normal part of the gut microbiome, they can also lead to serious infections, particularly in hospitalized patients who have received antibiotic treatment and have undergone changes in their intestinal microbiota [1].

Enterococci can acquire antibiotic resistance through mobile genetic elements (MGEs), plasmids, and transposons [2]. As a consequence, *E. faecium* is often resistant to multiple antibiotics, including ampicillin and vancomycin [3]. Vancomycin-resistant *E. faecium* (VRE*faecium*) is classified as a high priority pathogen by the WHO [4]. Vancomycin resistance is due to the presence of the *van*A gene, which is part of the *van*A operon [5] that is often carried by transposons such as Tn1546 [6].

In recent years, there has been a noticeable increase in infections caused by antibiotic-resistant enterococci. Specifically, *E. faecalis* has shown resistance to β-lactams, aminoglycosides, and fluoroquinolones, while *E. faecium* exhibits resistance to β-lactams, aminoglycosides, and glycopeptides, mainly due to the presence of the *van*A and *van*B genes [7]. In Italy, the number of reported VRE cases continues to rise. The latest ECDC report indicated that the percentage of VRE*faecium* ranged between 25% and 50% in 2023 [8]. In addition to VRE strains, *van*A-positive enterococci that show phenotypic susceptibility to vancomycin are known as vancomycin-variable enterococci (VVE). These VVE strains have been reported in Asia [9,10], Canada [11], Norway [12], and Denmark [13], and more recently, in India [14], Bangladesh [15] and Australia [16].

Both in vivo and in vitro studies indicate that VVE can revert to the VRE phenotype after vancomycin exposure, while the role of other antibiotics in inducing the same effect remains unclear [6,17,18]. The main concern is that due to their phenotypic susceptibility to vancomycin, these strains may escape traditional VRE detection methods, leading to underdiagnosis and silent transmission in healthcare settings where antibiotics are largely used, potentially causing hospital outbreaks of VRE strains [11].

However, most reports on the prevalence of VRE among *van*A-positive enterococci are limited to single institutions, and few multi-center or region-wide studies are available [7,19].

To prevent transmission of VRE, screening of patients upon hospital admission is of paramount importance. Rectal swabs are commonly used for VRE screening to detect carriers, prevent transmission and identify patients who may benefit from empiric VRE treatment in cases of enterococcal bacteremia [7]. The two main approaches used to detect VRE from rectal swabs are the traditional culture technique and molecular assays (MAs). While MAs are generally faster and more accurate, they tend to be more expensive and can detect the *van*A/*van*B target genes from non-viable VRE isolates; this is a well-known limitation of MAs that can potentially lead to apparently discordant results that can be difficult to interpret [7].

With the aim of obtaining a snapshot of the current prevalence of VRE colonization and identifying hotspots and critical areas, we conducted a point prevalence survey (PPS) within our hospital. We performed genomic analysis on the VRE isolates identified during the PPS to study their genetic diversity and transmission routes in an attempt to understand how these bacteria spread and evolved within the hospital. Overall, the objective of our study was to improve our understanding of VRE epidemiology and support more effective infection control strategies.

## 2. Results

### 2.1. Collection and Phenotypic Characterization of Isolates

Of 160 rectal swabs collected from consecutive patients, 54 (33.7%; 95% confidence interval (CI) 26.42–41.08%.) were positive for *van*A and/or *van*B by MA; 47 of these (29.3% of the total; 95% CI 22.32–36.43%) yielded positive cultures: 44 were identified as *E. faecium* and 3 as *E. faecalis*. The present study focuses on these 47 strains. Notably, of the 47 VRE patients positive at rectal screening, 23 (23/47; 49% with 95% CI 34.64–63.23%) were colonized on admission. The isolates were classified as VRE in 35 cases (35/47, 74.5%) and as VVE in 12 cases (12/47, 25.5%).

Isolates were tested for ampicillin, imipenem, linezolid, daptomycin, teicoplanin, tigecycline, vancomycin, and high-level gentamicin resistance. The results showed that 43 strains were resistant to ampicillin (43/47, 91.5%), while 34 of the 35 VRE strains were also resistant to teicoplanin. Additionally, 40 isolates (40/47 isolates, 85.1%) were resistant to high-level gentamicin. All the isolates were sensitive to tigecycline, while only one was resistant to linezolid (1/47 isolates, 2%). Imipenem susceptibility was tested on the *E. faecalis* isolates, and all of them were sensitive.

The distribution of the minimum inhibitory concentration (MIC) values for daptomycin ranged between 2 and ≥4, with a high prevalence of MICs >4 in both *E. faecium* and *E. faecalis*. These findings are summarized in Table 1. As expected, all the VVE strains were sensitive to both vancomycin and teicoplanin, while among the VRE strains, only TRCIO_21 showed susceptibility to teicoplanin.

### 2.2. Whole-Genome Sequencing (WGS) Analysis and Multilocus (ML) Phylogenetic Tree

All the strains included in this study were analyzed using molecular techniques, specifically whole-genome sequencing (WGS) and core-genome multilocus sequence typing (cgMLST).

The complete list of the strains, together with their total number of reads, QUAST assembly statistics, genome BUSCO completeness and annotations, is provided in Appendix A.

The QUAST assembly statistics indicated an average GC content of 38.09%, consistent with the species, and showed no undetermined bases. The average nucleotide identity (ANI) analysis demonstrated high similarity among most strains (over 99%), clustering them together (including the reference SRR24 strain). Three strains (TRCIO_17, 33, and 39) formed a distinct cluster, showing an ANI ranging from 88% to 90%, which supports the classification of all three as *E. faecalis* (Appendix A and Appendix A).

Appendix A shows the multilocus (ML) phylogenetic tree of the isolates included in our study. Out of the total, 13 isolates (13/47, 27.6%) came from patients who were already colonized when they were transferred from different hospitals in the Lazio region, while 24 isolates (24/47, 51%) were grown from patients who had been colonized in our institution 48 h after admission.

All the isolates shared the *van*HAX operon, except for five VVE strains: TRCIO_17, 30, 33, 37, and 39.

All the VRE strains possessed the complete two-component system (*van*R-*van*S), except TRCIO_35. Additionally, an insertion sequence (IS) labeled IS1251 was discovered in the intergenic region between *van*S and *van*H in strains TRCIO_04, 12, and 22. The same IS1251 insertion was found in 20 further strains, although in different locations.

Furthermore, eleven of the remaining isolates (TRCIO_06, 09, 15, 19, 23, 24, 25, 26, 34, 36, and 43) were found to contain an unidentified IS in the *van*S–*van*H intergenic region in addition to the IS1216V localized in Tn1546. No ISs were found in TRCIO_42. Interestingly, all the vancomycin-variable *E. faecalis* (VVE*faecalis*) isolates carried only the *van*T gene, which is associated with glycopeptide resistance [22]. In contrast, the vancomycin-variable *E. faecium* (VVE*faecium*) isolates lacking the *van*HAX operon contained the *van*Y gene, which is also responsible for glycopeptide resistance [23] (Appendix A).

The ResFinder analysis of the 47 genomes revealed complex resistomes, highlighting the presence of multiple acquired antibiotic resistance genes (Figure 1A,B).

Resistance to macrolides, streptogramin A and lincosamide is conferred by *erm(A)*, *erm(B)*, *erm(T)*, *Inu(B)*, *Inu(G)*, *Isa(A)*, *Isa(E)*, *msr(C)* and *vat-E*. The *msr(C)* and *erm(B)* genes were present in all the isolates except for three VVE strains (TRCIO_17, 30, and 33); TRCIO_39 only possessed *erm(B)*. Additionally, the *erm(T)* gene was found exclusively in TRCIO_6, 9, and 36, while the *Inu(B)* gene was present only in TRCIO_37. The *Isa(A)* and *Isa(E)* genes were identified in TRCIO_17, 33 and 39, and TRCIO_37, respectively (Figure 1 Panel B and Appendix A).

Except for TRCIO_12, 15, 16, 23, 24, 25, 26, 34, and 43, all the isolates exhibited tetracycline-associated resistance genes [*tet(M) and tet(L)*] (Figure 1A and Appendix A). A few isolates (TRCIO_17, 30, 31, 33, and 39) displayed only the *tet(M)* resistance gene (Figure 1B and Appendix A).

Of the total isolates, 15 (TRCIO_6, 9, 17, 19, 30, 31, 32, 33, 34, 36, 37, 39, 41, 42, and 44) showed no resistance to folate (*dfrG*). Phenicol resistance, which is encoded by *poxtA* and *cat* genes, was present in TRCIO_37 and TRCIO_22, respectively. This evidence is in contrast to the phenotypic results, which showed that only TRCIO_28 was resistant to linezolid. This discrepancy highlights the presence of non-expressed genes. All the isolates demonstrated resistance to aminoglycosides, supported by the presence of the following genes: *aac(6′)-Ii*, *ant(6)-Ia*, *aph(3′)-III*, *and aac(6′)-aph(2″)* (Figure 1A,B, Appendix A).

Analysis of the virulome revealed the presence of 25 virulence genes belonging to different categories [24] (Figure 2). Interestingly, some of these virulence genes are responsible for progression to endocarditis through biofilm formation: *empA*, *empB*, *empC* and *bepA* [24]. Two of the VVE*faecalis* isolates lacked virulence genes (TRCIO_17, and 39), while TRCIO_33 possessed only the *fms*15 gene.

In addition, all the isolates shared the virulence factor IS16, except for all the *E. faecalis* strains and two VVE*faecium* (TRCIO_30, and 37), which were the only two strains possessing the *van*Y gene.

PlasmidFinder revealed the presence of 16 plasmids shared among the isolates, as shown in Figure 3: RepUS43, RepUS15, RepUS12, Rep17, Rep11a, Rep2, Rep14a, Rep14b, Rep18a, Rep18b, Rep29, RepUS1, Rep6, Rep9a, Rep9b, and RepUS11.

Each plasmid carries resistance genes related to different antimicrobial agents, as summarized in Figure 4 and Appendix A.

Figure 4 shows that the plasmids contained both aminoglycoside-resistant genes [*aadA5*, *aadD*, *aph(2″)-Ia*, *aph(2″)-Ie*, *aph(3′)-III*, *aac(6′)-aph(2″)*, *aac(6′)-Ii*, *ant(6)-Ia*, *ant(9)-Ia*] and macrolide–lincosamide–streptogramin resistance genes [*erm(A)*, *erm(B)*, *erm(T)*, *Inu(B)*, *Inu(G)*, *Isa(E)*, *vat-E*]. However, fewer genes were found in Rep6 [*aph(3′)-III*, *ant(6)-Ia*] and Rep9 [*erm(B)*, *Isa(A)*], respectively.

Also, tetracycline resistance genes [*tet(45)*, *tet(L)*, *tet(M)*, *tet(S)*, *tet(O)*, *tet(W*)] were present in all the plasmids. On the contrary, the *bla*Z β-lactam gene was present only in a few plasmids (RepUS43, RepUS15, Rep17 and Rep2), while *dfrG*, which is responsible for the folate pathway inhibitors, was absent in Rep6. Additionally, RepUS1, Rep6, and Rep9b showed no phenicol resistance genes (*cat*, *fexB*, *poxtA*).

Interestingly, some plasmids (RepUS1, Rep6, Rep9a, Rep9b, and RepUS11) carried no glycopeptide resistance genes (*van*A, *van*B), while Rep6 was the only one showing the *van*X, *vat-D*, and *tet(K)* resistance genes, compared with the other plasmids.

TRCIO_15 and 43 were the only two strains presenting the same plasmid profile (RepUS15, and Rep17 only). TRCIO_19 was the only VRE isolate containing the Rep18a plasmid, while Rep2 was present only in three VRE isolates (TRCIO_12, 22, and 42) (Appendix A).

TRCIO_32 and 44 were the only two VVE isolates carrying the Rep14a plasmid. Notably, TRCIO_17 and TRCIO_39 were the only two strains that showed the same plasmid profile (RepUS43, and Rep9b) and no virulence genes. The Rep18a plasmid was found only in one VVE strain (TRCIO_16), while Rep2 was only in three (TRCIO_30, 33, and 37) and RepUS1 in one (TRCIO_37). The Rep6 plasmid, which is commonly found in *E. faecalis* strains, was present only in TRCIO_33, which was identified as an *E. faecalis* isolate (Appendix A).

### 2.3. cgMLST Analysis

The cgMLST analysis, which determined the clonal relationship using a maximum of 15 difference alleles as a reference threshold for clustering, identified seven clusters (Appendix A).

It is noteworthy that the strains that cluster in cluster types (CTs) CT-1, CT-2, CT-3, and CT-6 all belong to ST80, whereas those forming CT-5 and CT-7 are all ST1478. CT-4, in contrast, does not contain defined STs. Additionally, ten isolates do not cluster with one another or with other *E. faecium* isolates. Of these, four were identified as ST80, two as ST117, and three as ST971, 1126 and 1478, respectively.

In addition, four isolates (TRCIO_15, 17, 33, and 39) could not be identified as a specific ST and were categorized as “No group assigned” (Appendix A). The cgMLST analysis revealed that these isolates were lacking two housekeeping genes: *pstS* and *purK*. Among these isolates, three were identified as *E. faecalis* (TRCIO_17, 33, and 39) and were closely related to one another, but they were distinct from the *E. faecium* strains (Appendix A). The cgMLST analysis also indicated that CT-1 was the most prevalent, grouping 42.5% of the isolates, and was found in all the wards except the ICU. CT-1 was mostly prevalent across all the hospital wards, containing 25% (5/20) of isolates found in W-1, and 25% (5/20) in W-2. In contrast, the totality of the CT-2 isolates came from W1. Isolates from the ICU clustered only in CT-6 and CT-7 (50% and 20%, respectively) (Figure 5 and Appendix A).

## 3. Discussion

Enterococci are part of the normal gut microbiota in healthy humans and animals, but they can enter the environment through fecal contamination [25]. These bacteria are involved in a wide range of infections and are classified as human pathogens [1]. The rise in VRE infections in hospitals is particularly alarming; VRE enterococci are responsible for a substantial number of healthcare-associated infections and are linked to high mortality rates, especially among patients with pre-existing health conditions [26,27]. While some studies have reported increased mortality associated with VRE infections, others suggested that confounding factors, such as disease severity and population selection, might influence such findings [28]. In a previous work, we reported that in our hospital setting, which is a referral center that caters for patients with infectious diseases, those with hospital-acquired VRE*faecium* rectal colonization faced a higher risk of VRE*faecium* infections compared to those without such colonization [7]. Interestingly, our PPS showed that 27.6% of positive patients were already colonized prior to admission, indicating the wide spread and circulation of these strains.

Our study aimed at characterizing at the molecular level all the VRE and VVE strains collected during a one-day PPS in order to gain a snapshot of the prevalence and circulation within our hospital of potentially hazardous strains carrying genes linked with antibiotic resistance and virulence (Table 1 and Figure 1A,B).

Vancomycin resistance is linked with the presence of the *van*A gene, which is part of the *van*A operon (often carried by transposons such as Tn1546) and consists of the following main components: the regulatory region (*van*R), a membrane-bound histidine kinase glycopeptide sensor (*van*S), the glycopeptide resistance region (*van*HAX), and a terminal region containing accessory elements (*van*Y, *van*Z). In particular, the emergence of resistance is due to the presence of the two-component *van*R*–van*S system, which is responsible for constitutive glycopeptide resistance [5].

VVE strains are bacteria that possess the *van*A gene but display a vancomycin-susceptible phenotype, as they lack the *van*R–*van*S genes; however, these strains can develop permanent drug resistance upon exposure to glycopeptides. VVE may therefore have important clinical implications due to their potential to revert to a VRE phenotype following antibiotic exposure. Because of their vancomycin-susceptible phenotype, VVE strains often go undetected in traditional VRE control programs, which are typically based on the use of vancomycin-containing selective media. As a result, these isolates may silently spread within healthcare settings [29], where the transposon-mediated horizontal transfer of the *van*A gene may further exacerbate the risk of dissemination.

The results obtained from the VVE strains in our study were consistent with the susceptible phenotype reported in the literature, as all the strains lacked the *van*R gene. Some of these strains did contain the complete *van*HAX operon or the *van*Y gene, which also confers glycopeptide resistance (Appendix A). The three *E. faecalis* strains examined (TRCIO_17, 33, and 39) also lacked the *van*R gene and only possessed the *van*T gene, which is also associated with glycopeptide resistance [5].

Among the possible mechanisms of vancomycin resistance in VRE strains are deletions and/or a point mutation (C⟶A at position 5727) in *van*S [5,30], or previous treatments with drugs such as ciprofloxacin [17], ceftriaxone [6], and daptomycin [18]. In our VRE population, the *van*R*–van*S system was detected in all the isolates, as expected, except for TRCIO_35. Similar cases of VRE*faecium* lacking the *van*R–*van*S two-component system are described in the literature [5]; in these strains, vancomycin resistance may result from an IS (IS1251) in the *van*S–*van*H intergenic region [31], as is the case for TRCIO_04, 12 and 22. In TRCIO_35, the IS1251 sequence was not found in this intergenic region, which makes this particular strain worthy of further studies to understand how its resistant phenotype originated. In the remaining eleven VRE strains, a different IS (IS1216V) was found in Tn1546. Both ISs are associated with glycopeptide resistance and *van*S disruption [32], which makes our data consistent with previous studies.

Although enterococci have long been considered bacteria with little or no virulence, since the emergence of VRE strains several virulence factors have been identified [33]. We therefore explored the presence of genes associated with virulence in our isolates. We detected the same virulence genes in VVE*faecium* strains as in VRE*faecium* (Figure 2), suggesting the potential pathogenicity of VVE strains in case of phenotypic reversion. Our results show that all the analyzed strains had a similar virulence profile; two of the VVE*faecalis* strains (TRCIO_17, and 39) showed no virulence determinants, while the remaining one (TRCIO_33) only had *fms*15 (surface-exposed cell-wall-anchored protein) (Figure 2 and Appendix A). These results are in contrast to the findings of Georges et al. [34], probably due to the small number of *E. faecalis* isolates included in our study. Most isolates (VRE and VVE), with the exception of two VVE*faecium* (TRCIO_30, and 37) and all three *E. faecalis* isolates, share the presence of the virulence factor IS16, which has been described in invasive hospital *E. faecium* strains [35]. In the *E. faecalis* strains, the absence of IS16 is consistent with their low virulence and the absence of the *van*HAX operon, while in the two VVE*faecium* strains, it could be explained by the absence of the *van*HAX operon (Appendix A).

Figure 3 and Appendix A show the distribution of plasmids associated with antibiotic resistance found in our strains; it is noteworthy that the majority of VRE isolates share the presence of the same plasmids (RepUS43, RepUS15, RepUS12, Rep17, and Rep11a). The majority of the VVE isolates also shared the same plasmids as the VRE strains (RepUS43, RepUS12, RepUS15, Rep17, and Rep11a), but they also carried a more heterogenous variety of plasmids, some unique to VVE strains (RepUS1, Rep6, Rep9a, Rep9b, and Rep11s). These results highlight the potential threat posed by VVE strains, as they may represent a reservoir of plasmids carrying virulence and resistance genes, potentially contributing to the spread of enterococci carrying virulence or antibiotic resistance genes, or both.

Genomic analysis also allowed us to determine the STs of the strains and their association within CTs across the hospital wards. The predominant ST was ST80, which has been widely reported to be prevalent not only in Italy [36] but also across Europe [37] and worldwide [38].

Other STs detected, to a lesser extent, were ST117, ST971, ST1126, and ST1478. Four isolates (the three *E. faecalis* and one *E. faecium* (TRCIO_15)) could not be assigned to a defined ST (Appendix A), possibly due to the absence of the *purK* and *pstS* housekeeping genes in these isolates. While the absence of the *pstS* gene may be attributed to the rapid dissemination of untypable enterococci over the years [39], there is currently no literature regarding the absence of the *purK* gene and this could be the focus of further research. Moreover, cgMLST analysis showed that in almost all the CTs (CT-1, 2, 3, 5, 6 and 7), there was a homogeneity of STs (ST80 and ST1478), while CT-4 was composed only of *E. faecalis* strains.

When we analyzed the distribution of CTs across the hospital wards, we found that CT-1 was the most prevalent, comprising 42.5% of the strains, and was found in all the wards except the ICU, while CT-2 was the most prevalent in W-1, including 100% of the isolates. The strains from the ICU clustered only in CT-6 and CT-7 (Figure 5).

It is interesting to note that CT-2 and CT-5 account for 100% of the isolates in W-1, suggesting intra-ward transmission of the strains; on the other hand, the remaining CTs (CT-1, CT-3, CT-4, CT-6, and CT-7) contained isolates deriving from different wards.

The strains collected for our study were grown from rectal swabs taken at admission and therefore provided a picture of the colonization of patients rather than infection. Despite the limitations of research conducted in a single center and on a relatively small number of isolates collected during a PPS, we believe that our results provide a snapshot that is significant in highlighting the importance of active surveillance in hospital settings to identify potential sources of outbreaks.

## 4. Materials and Methods

### 4.1. Hospital Setting and PPS

The National Institute for Infectious Diseases “Lazzaro Spallanzani” is a 160-bed referral hospital specializing in infectious diseases. It is a unique scientific institute dedicated to the hospitalization and care of infectious disease patients. A PPS was conducted on all the patients to define the extent of VRE colonization and to characterize the obtained isolates through WGS.

### 4.2. Rectal Sampling and Phenotypic Characterization of Isolates

Rectal swabs were collected on 06/05/2024 from 160 patients using fecal E-swabs^®^ (Copan; Brescia, Italy). Samples were analyzed using the Allplex™ Entero-DR Assay Kit (Seegene Inc., Seoul, Republic of Korea); those that tested positive for *van*A and/or *van*B resistance genes were cultured using CHROMID^®^ VRE Agar (bioMérieux Italia, Florence, Italy). A VVE strain was suspected when the culture of a *van*A- or *van*B-positive sample was negative on the selective medium; the sample was therefore plated on Columbia CNA Agar (Thermo Fisher Scientific, Waltham, MA, USA). The grown isolate was then confirmed as VVE after confirming the presence of the *van*A gene through MA and WGS. The agar plates were incubated at 37 °C under aerobic conditions, and the growing colonies were identified using the MALDI-TOF Biotyper Sirius System (Bruker Daltonics, Bremen, Germany) and MBT Compass software (version 4.2). Antimicrobial susceptibility testing (AST) of the isolates was subsequently performed via Phoenix panels (PMIC96) run on the PMICR 96 Phoenix system (Becton Dickinson Diagnostics, San Jose, CA, USA). The MICs were interpreted following the recent guidelines established by the EUCAST [40]. The whole workflow is summarized in Appendix A.

### 4.3. Whole-Genome Sequencing, Multilocus (ML) Phylogenetic Tree and Core-Genome Multilocus Sequence Typing (cgMLST) Analysis

WGS was performed on all the samples that showed positive growth by culture (47 strains). Libraries were prepared using the Nextera DNA Library Prep Kit, and sequencing was performed using the MiSeq sequencer (Illumina, Inc. San Diego, CA, USA). The sequences obtained were preliminarily processed using the Illumina BaseSpace system and converted into FASTQ files for subsequent bioinformatics analysis.

The raw reads’ quality was evaluated according to Andrews [41], followed by de novo assembly using Unicycler (version 0.5) with default parameters [42]. The assembly statistics were checked using QUAST (version 5.0) [43], while the genome completeness was assessed using BUSCO (version 5.4.7) [44].

#### 4.3.1. Assembly and Annotations

The assembled genomes were then annotated with Prokka (version 1.14.5) [45] and used to check for the presence of the *van*A cluster genes related to the Tn1546 transposon using BLAST+ (version 2.12.0). The alignment of the short reads from our isolates was performed with Bowtie2 (version 2.5.4) on reference sequences obtained from NCBI (GeneBank: KR047792, M97297). The .sam files obtained were converted into .bam files, then sorted and indexed using Samtools (version 1.21). The display of the indexed .bam file was managed with the IGV software (version 2.19.2) to check for insertions.

#### 4.3.2. Pangenomes

Pangenomes were inferred using Roary (version 3.13.0) [46], and an ML phylogenetic tree was built on the identified core genes using raxmlHPC (version 8.2.12) set with the GTRGAMMAI algorithm as a substitution model and 1000 bootstraps [47]. The ML tree was visualized in FigTree (version 1.4.4) (available at http://tree.bio.ed.ac.uk/software/figtree/, accessed on 12 January 2025), rooted in the reference *E. faecium* SSR24 strain, and further edited with Inkscape (version 0.92) (available at https://inkscape.org, accessed on 12 January 2025). The ANI was calculated based on the entire genomes with MUMMER (version 3.1) within the pyANI script [48]; in particular, pyANI was used to evaluate the overall percentage of identity of the strains, including the reference, using the ANIm method. 

#### 4.3.3. AMR Gene Analysis

Antimicrobial resistance (AMR) features were identified separately with the Resistance Gene Identifier (RGI) (version 5.2.0) [49] docker container of the Comprehensive Antibiotic Resistance Database (CARD) (version 3.1.4) (available at https://card.mcmaster.ca/, accessed on 12 January 2025) and with the ResFinder webserver of the Center for Genomic Epidemiology (CGE) (available at http://genepi.food.dtu.dk/resfinder, accessed on 17 January 2025). The threshold for a minimum percentage of sequence identity was established at 100%, with a required alignment length of >98%.

#### 4.3.4. Pathogenicity and Virulence

The prediction of the pathogenicity of a bacterium toward the human host was obtained using the PathogenFinder online tool (version 1.1) (available at https://cge.food.dtu.dk/services/PathogenFinder/, accessed on 14 February 2025), while the characterization of plasmid sequences in the WGS was performed using PlasmidFinder (version 2.1) (available at https://cge.food.dtu.dk/services/PlasmidFinder/, accessed on 14 February 2025). The virulence genes in bacterial genomes of interest were identified using the VirulenceFinder online tool (version 2.0) (available at https://cge.food.dtu.dk/services/VirulenceFinder/, accessed on 17 March 2025). On the other hand, the MGEs and their relationship to antimicrobial resistance genes and virulence factors were identified using the Mobile Genetic Elements tool of the CGE (available at https://cge.food.dtu.dk/services/MobileElementFinder/, accessed on 17 March 2025).

Additional MGEs were predicted using the MOB-suite software docker container (version 3.0.3) (available at https://github.com/phac-nml/mob-suite, accessed on 19 March 2025), while promoter sequences were obtained from BPROM (available at http://www.softberry.com/, accessed on 19 March 2025).

#### 4.3.5. Phylogenetic Analysis

To further explore the genetic relationships with the identification of the STs and CTs, we used the WGS-based cgMLST scheme (version 1.0), employing the Ridom SeqSphere+ software (Ridom GmbH, Münster, Germany) with default settings. A cgMLST, consisting of 1423 genes and built on the reference genome ID GenBank NCBI-ID-CP038996, was identified through a standardized procedure reported in the software manual and analyzed to construct a minimum spanning tree (MST), based on the respective genetic distances. Based on the allelic differences of the genes, the clonal relationship was determined using a maximum of 15 allelic differences as a reference threshold for clustering. The retrieved MLST sequences of the 47 strains were aligned with MUSCLE (version 3.8.31) [50].

All the raw reads generated were submitted to the Sequence Read Archive (SRA) under the BioProject ID PRJNA1243017.

### 4.4. Statistical Evaluation

Descriptive statistics were used to summarize the prevalence of VRE and VVE among the screened patients. The proportions of positive cases were expressed as percentages with the corresponding 95% CIs, calculated using the Wilson score method. These CIs provide an estimate of the precision of the observed proportions, which is particularly suitable for small sample sizes. No inferential statistical tests were applied, as this study was designed to provide a descriptive snapshot of the colonization and resistance patterns within a single-day PPS. All the calculations were performed using Microsoft Excel (version 2305).

## 5. Conclusions

In conclusion, this study provides a comprehensive analysis of VRE and VVE within a hospital setting. The findings highlight the significant prevalence of VRE and VVE strains, with a notable portion of patients already colonized upon admission (13/47; 27.6%). Genomic analysis revealed multiple STs and CTs, with ST80 being the most prevalent. The presence of various resistance genes and virulence factors was confirmed, underscoring the potential pathogenicity of these strains. The identification of specific resistance genes and mutations through WGS can help refine antibiotic treatment protocols and enhance infection control measures. Integrating genomic data with clinical information will improve the ability to predict and prevent future VRE infections. While future research should aim to address these limitations, the fact that none of the patients considered in our study developed a VRE bloodstream infection during the 30-day observation period after the first positive rectal swab highlights the importance of constant and effective surveillance in support of infection control measures.

Our study describes a surveillance program based on phenotypic as well as molecular data. While relying solely on culture-based methods may lead to an underestimation of colonization, particularly in cases of low bacterial load, MA on the other hand offers higher sensitivity and can detect resistance genes even in the absence of viable organisms. The potential for discrepancies or results that are difficult to interpret can be overcome by integrating genomic approaches into routine surveillance protocols, whenever possible.

A key aspect of an effective surveillance program should be the integration of targeted MA into routine microbiological diagnostics, to enable early detection of resistance determinants, and the improvement of antimicrobial stewardship policies. Further studies involving larger, multi-center studies are needed to understand the origin and evolution of vancomycin resistance in enterococci. Comparing the genomes of human enterococci with those from animal and environmental sources would help understand the factors underlying the widespread emergence of VRE that currently represent an important challenge in the fight against the rise of antibiotic resistance.

## Figures and Tables

**Figure 1 antibiotics-14-00795-f001:**
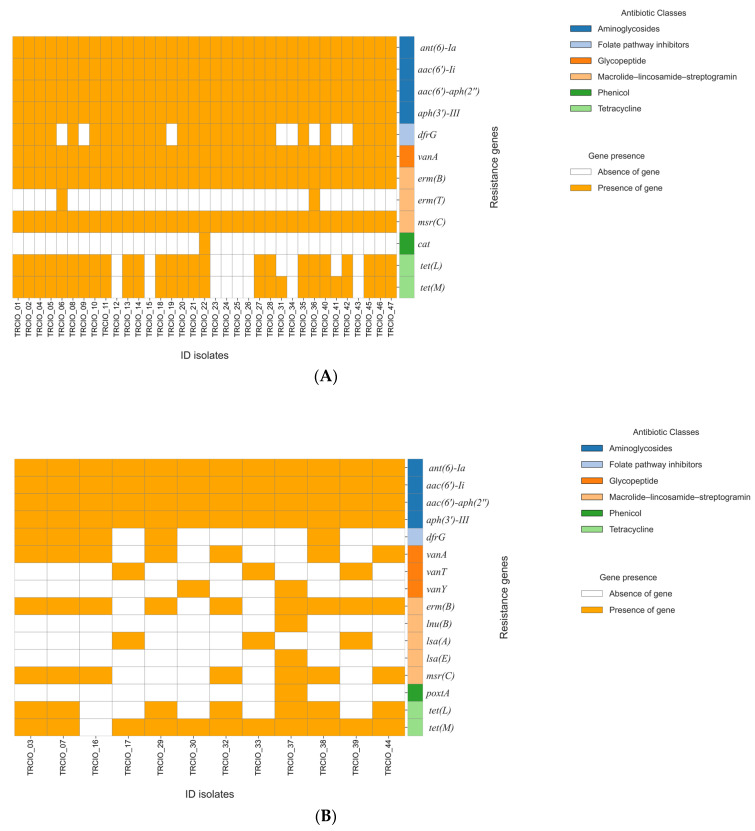
Molecular analysis of AMR genes performed on VRE and VVE isolates. Panel (**A**): molecular characterization of vancomycin-resistant enterococci (VRE) isolates; and Panel (**B**): molecular characterization of vancomycin-variable enterococci (VVE) isolates.

**Figure 2 antibiotics-14-00795-f002:**
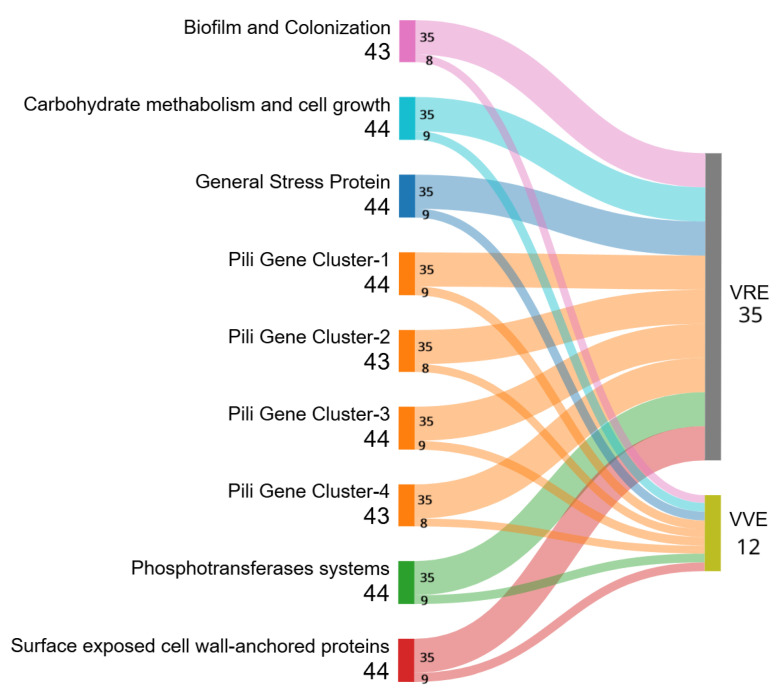
Analysis of virulence determinants performed on 35 VRE and 12 VVE strains. A full list of the detected virulence genes can be found in Appendix A. The width of each flow represents the number of isolates carrying a specific virulence gene, highlighting shared and unique profiles between the two groups. Figure created through SankeyMATIC (available at https://sankeymatic.com/, accessed on 16 April 2025).

**Figure 3 antibiotics-14-00795-f003:**
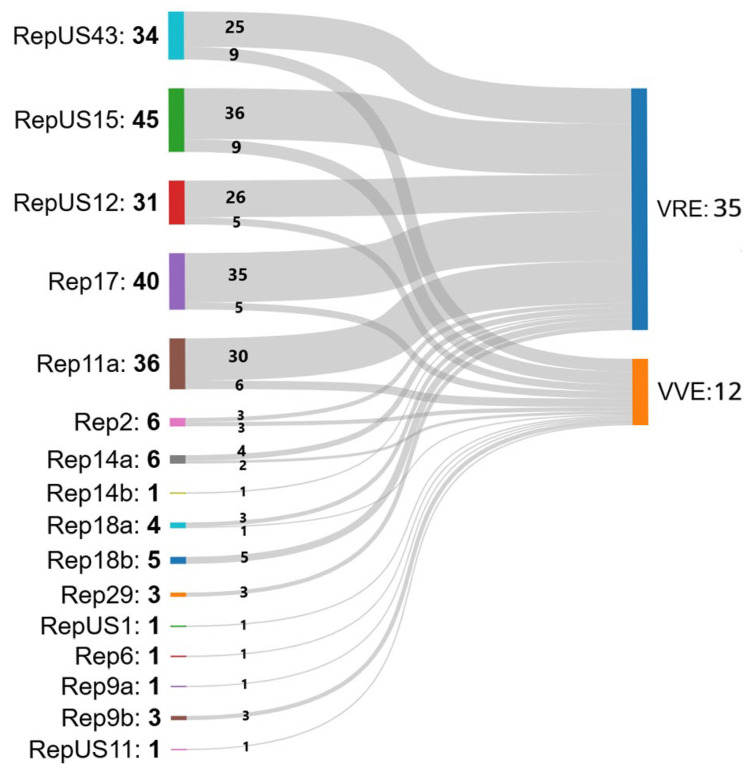
Distribution of plasmids in 35 VRE and 12 VVE strains. Distribution of plasmids among vancomycin-resistant enterococci (VRE) and vancomycin-variable enterococci (VVE) isolates. The width of each flow represents the number of isolates carrying a specific plasmid, highlighting shared and unique plasmid profiles between the two groups. Figure created through SankeyMATIC (available at https://sankeymatic.com/, accessed on 16 April 2025).

**Figure 4 antibiotics-14-00795-f004:**
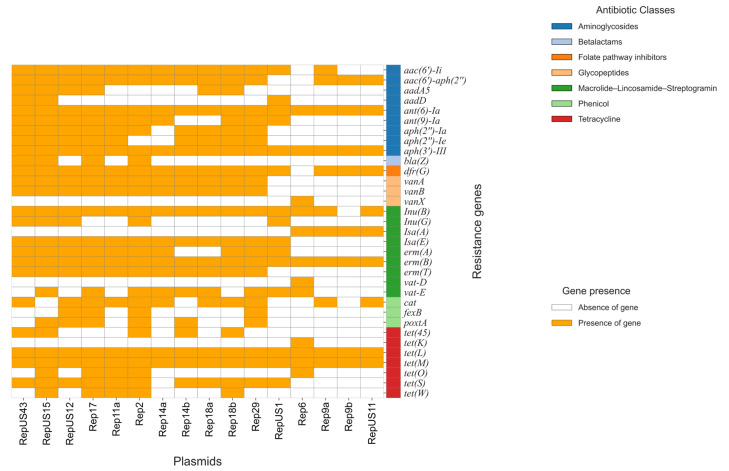
Resistance genes present in various plasmids carried by all the study isolates.

**Figure 5 antibiotics-14-00795-f005:**
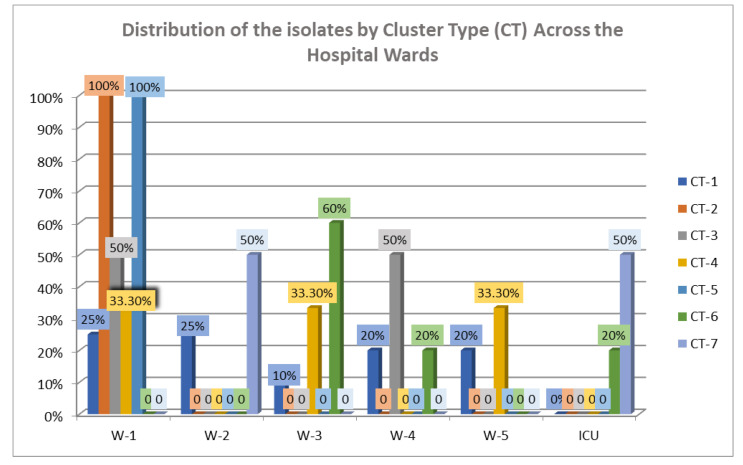
Distribution of the isolates by cluster type (CT) across hospital wards. W (1–5): W-1: Hepatology Unit, W-2: Viral Immunodeficiency Unit, W-3: Immune Systemic Infections Unit, W-4: Respiratory System Infectious Diseases Unit, W-5: High-Intensity Care Infectious Disease Unit; ICU: Intensive Care Unit.

**Table 1 antibiotics-14-00795-t001:** Antimicrobial susceptibility results for the clinical isolates (all grown from rectal swabs).

ID Strain	Organism ^2^	Type ^3^	Date	Antibiotic MIC Values (mg/L) ^1^							
				AMP	IMP ^4^	LZD	DAP ^5^	TEC	TG	VAN	GEN-hl
**TRCIO_01**	*E. faecium*	VRE	6 May 2024	>16, R		2, S	>4	>8, R	0.25, S	>8, R	>500, R
**TRCIO_02**	*E. faecium*	VRE	16 April, 2024	>16, R		2, S	>4	>8, R	0.25, S	>8, R	>500, R
**TRCIO_03**	*E. faecium*	VVE	6 May 2024	>16, R		2, S	4	≤0.5, S	0.25, S	≤0.5, S	>500, R
**TRCIO_04**	*E. faecium*	VRE	17 April 2024	>16, R		2, S	>4	>8, R	0.25, S	>8, R	>500, R
**TRCIO_05**	*E. faecium*	VRE	18 April 2024	>16, R		2, S	>4	>8, R	0.25, S	>8, R	>500, R
**TRCIO_06**	*E. faecium*	VRE	3 May 2024	>16, R		2, S	4	>8, R	0.125, S	>8, R	>500, R
**TRCIO_07**	*E. faecium*	VVE	6 May 2024	>16, R		2, S	>4	≤0.5, S	0.25, S	≤0.5, S	>500, R
**TRCIO_08**	*E. faecium*	VRE	6 May 2024	>16, R		2, S	>4	>8, R	0.25, S	>8, R	>500, R
**TRCIO_09**	*E. faecium*	VRE	3 April 2024	>16, R		4, S	4	>8, R	≤0.125, S	>8, R	>500, R
**TRCIO_10**	*E. faecium*	VRE	18 April 2024	>16, R		2, S	>4	>8, R	0.25, S	>8, R	>500, R
**TRCIO_11**	*E. faecium*	VRE	6 May 2024	>16, R		2, S	>4	>8, R	0.25, S	>8, R	>500, R
**TRCIO_12**	*E. faecium*	VRE	6 May 2024	>16, R		4, S	4	>8, R	0.125, S	>8, R	>500, R
**TRCIO_13**	*E. faecium*	VRE	6 May 2024	>16, R		2, S	>4	>8, R	0.25, S	>8, R	>500, R
**TRCIO_14**	*E. faecium*	VRE	7 April 2024	>16, R		2, S	4	>8, R	0.25, S	>8, R	>500, R
**TRCIO_15**	*E. faecium*	VRE	25 January 2024	>16, R		2, S	4	>8, R	≤0.125, S	8, R	>500, R
**TRCIO_16**	*E. faecium*	VVE	6 May 2024	>16, R		1, S	4	≤0.5, S	≤0.125, S	≤0.5, S	>500, R
**TRCIO_17**	*E. faecalis*	VVE	6 May 2024	2, S	2, S	1, S	4	≤0.5, S	≤0.125, S	1, S	≤500, S
**TRCIO_18**	*E. faecium*	VRE	6 May 2024	>16, R		4, S	>4	>8, R	0.25, S	>8, R	>500, R
**TRCIO_19**	*E. faecium*	VRE	6 May 2024	>16, R		2, S	4	>8, R	0.125, S	>8, R	>500, R
**TRCIO_20**	*E. faecium*	VRE	6 May 2024	>16, R		2, S	>4	>8, R	0.25, S	>8, R	>500, R
**TRCIO_21**	*E. faecium*	VRE	20 February 2024	>16, R		2, S	4	≤0.5, S	0.25, S	>8, R	>500, R
**TRCIO_22**	*E. faecium*	VRE	1 May 2024	>16, R		2, S	4	>8, R	0.25, S	>8, R	≤500, S
**TRCIO_23**	*E. faecium*	VRE	11 April 2024	>16, R		2, S	4	>8, R	≤0.125, S	>8, R	>500, R
**TRCIO_24**	*E. faecium*	VRE	26 April 2024	>16, R		2, S	2	>8, R	≤0.125, S	>8, R	≤500, S
**TRCIO_25**	*E. faecium*	VRE	14 January 2024	16, R		2, S	4	>8, R	≤0.125, S	>8, R	>500, R
**TRCIO_26**	*E. faecium*	VRE	4 May 2024	>16, R		2, S	4	>8, R	<0.125, S	>8, R	>500, R
**TRCIO_27**	*E. faecium*	VRE	6 May 2024	>16, R		2, S	>4	>8, R	0.25, S	>8, R	>500, R
**TRCIO_28**	*E. faecium*	VRE	27 April 2024	>16, R		>4, R	>4	>8, R	≤0.125, S	>8, R	>500, R
**TRCIO_29**	*E. faecium*	VVE	30 April 2024	>16, R		2, S	>4	≤0.5, S	≤0.25, S	≤0.5, S	>500, R
**TRCIO_30**	*E. faecium*	VVE	6 May 2024	4, S		1, S	>4	≤0.5, S	≤0.125, S	2, S	≤500, S
**TRCIO_31**	*E. faecium*	VRE	6 May 2024	>16, R		2, S	2	>8, R	0.125, S	>8, R	≤500, S
**TRCIO_32**	*E. faecium*	VVE	6 May 2024	>16, R		2, S	>4	≤0.5, S	0.125, S	1, S	>500, R
**TRCIO_33**	*E. faecalis*	VVE	6 May 2024	2, S	4, S	2, S	2	≤0.5, S	≤0.125, S	2, S	>500, R
**TRCIO_34**	*E. faecium*	VRE	6 May 2024	>16, R		4, S	>4	>8, R	0.125, S	>8, R	>500, R
**TRCIO_35**	*E. faecium*	VRE	6 May 2024	>16, R		2, S	4	>8, R	0.25, S	>8, R	>500, R
**TRCIO_36**	*E. faecium*	VRE	6 May 2024	>16, R		2, S	4	>8, R	0.25, S	>8, R	>500, R
**TRCIO_37**	*E. faecium*	VVE	6 May 2024	>16, R		2, S	4	≤0.5, S	0.125, S	2, S	>500, R
**TRCIO_38**	*E. faecium*	VVE	6 May 2024	>16, R		2, S	>4	≤0.5, S	≤0.125, S	≤0.5, S	>500, R
**TRCIO_39**	*E. faecalis*	VVE	3 May 2024	2, S	2, S	1, S	2	≤0.5, S	≤0.125, S	1, S	≤500, S
**TRCIO_40**	*E. faecium*	VRE	6 May 2024	>16, R		2, S	>4	>8, R	0.25, S	>8, R	>500, R
**TRCIO_41**	*E. faecium*	VRE	9 April 2024	>16, R		2, S	2	>8, R	0.125, S	>8, R	≤500, S
**TRCIO_42**	*E. faecium*	VRE	6 May 2024	>16, R		2, S	4	>8, R	≤0.125, S	>8, R	>500, R
**TRCIO_43**	*E. faecium*	VRE	25 December 2023	>16, R		4, S	4	>8, R	≤0.125, S	>8, R	>500, R
**TRCIO_44**	*E. faecium*	VVE	6 May 2024	>16, R		4, S	4	≤0.5, S	0.125, S	≤0.5, S	>500, R
**TRCIO_45**	*E. faecium*	VRE	24 April 2024	>16, R		2, S	>4	>8, R	0.25, S	>8, R	>500, R
**TRCIO_46**	*E. faecium*	VRE	6 May 2024	>16, R		2, S	>4	>8, R	0.25, S	>8, R	>500, R
**TRCIO_47**	*E. faecium*	VRE	6 May 2024	>16, R		2, S	>4	>8, R	0.125, S	>8, R	>500, R

^1^ MICs: minimum inhibitory concentrations; AMP, ampicillin; IPM, imipenem; LZD, linezolid; DAP, daptomycin; TEC, teicoplanin; TG, tigecycline; VAN, vancomycin; GEN-hl, gentamycin high-level; S, susceptible; R, resistant; ^2^ *E. faecium*: *Enterococcus faecium*; *E. faecalis*: *Enterococcus faecalis*; ^3^ VRE: vancomycin-resistant enterococci; VVE: vancomycin-variable enterococci; ^4^ IMP tested only for *E. faecalis*; ^5^ DAP: daptomycin is currently not approved by the EMA for treating enterococcal infections. The European Committee on Antimicrobial Susceptibility Testing (EUCAST) lists the daptomycin breakpoints for *Enterococcus* species as “IE” (insufficient evidence), and daptomycin is not recommended for therapy [20,21]. Text in red represents the molecules to which the isolates were resistant.

## Data Availability

The data can be found in the Excel database, created ad hoc and archived at the authors’ institution (INMI L. Spallanzani IRCCS, Rome, Italy).

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
