# Peer review of "Surveillance and Characterization of Vancomycin-Resistant and Vancomycin-Variable Enterococci in a Hospital Setting"

_antibiotics, 2025, doi:10.3390/antibiotics14080795_

Round 1

Reviewer 1 Report

Comments and Suggestions for Authors

Dear Editors,
after careful evaluation of the manuscript entitled:
"Surveillance and Characterization of Vancomycin-Resistant and Vancomycin-Variable Enterococci in a Hospital Setting",
I would like to point out that this is a fundamentally done and methodologically strong study that addresses an extremely important issue in the field of antimicrobial resistance in the clinical environment.
The authors carried out at a high technical level:
• Point Prevalence Survey (PPS) for VRE and VVE colonization of patients,
• molecular and genomic characterization (WGS, cgMLST) of isolates,
• identification of resistome, virulence and plasmid profile,
• and mapped potential routes of transmission within hospital wards.
I also highlight the significant new contribution that the work brings regarding VVE strains - their clinical significance, the possibility of phenotypic conversion and the risk of being neglected in clinical diagnostics.
The authors were asked to:
• further deepen the discussion about the clinical implications of VVE,
• clearly identify the limitations of one institution and point sample,
• propose strategies for the clinical implementation of genetic findings.
These additions will further increase the value of the work for a wide readership such as microbiologists as well as clinicians and managers of hospital infections.
Final recommendation
Considering:
• topicality of the topic,
• methodological consistency,
• clear practical and scientific relevance,
• as well as the quality of data presentation and analysis,
I recommend accepting the work with minor revisions (Minor Revision).

Author Response

Review Report Form

Open Review

( ) I would not like to sign my review report

(x) I would like to sign my review report

Quality of English Language

( ) The English could be improved to more clearly express the research.

(x) The English is fine and does not require any improvement.

Yes         Can be improved             Must be improved          Not applicable

Does the introduction provide sufficient background and include all relevant references?

(x)          ( )           ( )            ( )

Is the research design appropriate?

(x)          ( )           ( )            ( )

Are the methods adequately described?

(x)          ( )           ( )            ( )

Are the results clearly presented?

(x)          ( )           ( )            ( )

Are the conclusions supported by the results?

(x)          ( )           ( )            ( )

Are all figures and tables clear and well-presented?

(x)          ( )           ( )            ( )

Comments and Suggestions for Authors

Dear Editors,

after careful evaluation of the manuscript entitled:

"Surveillance and Characterization of Vancomycin-Resistant and Vancomycin-Variable Enterococci in a Hospital Setting",

I would like to point out that this is a fundamentally done and methodologically strong study that addresses an extremely important issue in the field of antimicrobial resistance in the clinical environment.

The authors carried out at a high technical level:

  • Point Prevalence Survey (PPS) for VRE and VVE colonization of patients,
  • molecular and genomic characterization (WGS, cgMLST) of isolates,
  • identification of resistome, virulence and plasmid profile,
  • and mapped potential routes of transmission within hospital wards.

I also highlight the significant new contribution that the work brings regarding VVE strains - their clinical significance, the possibility of phenotypic conversion and the risk of being neglected in clinical diagnostics.

The authors were asked to:

  • further deepen the discussion about the clinical implications of VVE,
  • clearly identify the limitations of one institution and point sample,
  • propose strategies for the clinical implementation of genetic findings.

These additions will further increase the value of the work for a wide readership such as microbiologists as well as clinicians and managers of hospital infections.

Final recommendation

Considering:

  • topicality of the topic,
  • methodological consistency,
  • clear practical and scientific relevance,
  • as well as the quality of data presentation and analysis,

I recommend accepting the work with minor revisions (Minor Revision).

Point by point responses

  1. “Further deepen the discussion about the clinical implications of VVE.”

Thank you for this valuable suggestion. We have modified the discussion section to further elaborate on the clinical implications of VVE, particularly focusing on its impact on treatment decisions, infection control strategies, and patient outcomes (page 13, lines 307-315).

  1. “Clearly identify the limitations of one institution and point sample.”

We have now explicitly acknowledged the limitations associated with conducting the study in a single institution and using a PPS. However, we also stressed that our Institute is also a regional referral centre that caters for patients coming from a wider geographical area (Page 13, lines 290-291, and page 15 lines 380-385).

  1. “Propose strategies for the clinical implementation of genetic findings.”

We appreciate the reviewer’s suggestion to address the clinical implementation of genetic findings.

We modified the conclusions paragraph to explain the challenges posed by MA in general and to suggest possible steps to improve the usefulness of genetic findings, when available: page 17, lines 501-514).

Reviewer 2 Report

Comments and Suggestions for Authors

The study effectively leverages modern molecular and genomic techniques to provide valuable insights into VRE and VVE epidemiology within their hospital. The findings are significant and contribute to the understanding of antibiotic resistance challenges.

  • While rectal colonization is a primary reservoir and a key indicator of transmission risk, it doesn't directly measure active infections. Patients can be colonized without developing disease. The study did follow up on bloodstream infections but the overall prevalence refers to colonization, not necessarily active infection rates. 

- Since this manuscrpt aims to address an issue that has potential of widespread attention  from various field of discipline, the below lissted issues would better be resolved in order to enhance the level of clarity for general readers that are outside the field of clinical microbiology. This will prevent readers from thinking that there is a logical error in their inferences.

Example: “Text 1 (Lines 85-86): States that MAs "can detect the vanA/vanB target genes from non-viable VRE isolates, potentially leading to discordant results”

>> Detecting genes from non-viable bacteria doesn't mean the patient has an active, culturable infection. The results therefore refer to a positive MA but a negative culture, which is precisely the VVE phenomenon where the gene is present but not expressed phenotypically. This is a known and accepted limitation/nuance of molecular diagnostics in clinical microbiology. From the perspective of general audience (readers), general informative basic molecular biology and microbiology information can be included.

Author Response

Review Report Form

Open Review

(x) I would not like to sign my review report

( ) I would like to sign my review report

Quality of English Language

( ) The English could be improved to more clearly express the research.

(x) The English is fine and does not require any improvement.

Yes         Can be improved             Must be improved          Not applicable

Does the introduction provide sufficient background and include all relevant references?

(x)          ( )           ( )            ( )

Is the research design appropriate?

(x)          ( )           ( )            ( )

Are the methods adequately described?

(x)          ( )           ( )            ( )

Are the results clearly presented?

(x)          ( )           ( )            ( )

Are the conclusions supported by the results?

(x)          ( )           ( )            ( )

Are all figures and tables clear and well-presented?

(x)          ( )           ( )            ( )

Comments and Suggestions for Authors

The study effectively leverages modern molecular and genomic techniques to provide valuable insights into VRE and VVE epidemiology within their hospital. The findings are significant and contribute to the understanding of antibiotic resistance challenges.

While rectal colonization is a primary reservoir and a key indicator of transmission risk, it doesn't directly measure active infections. Patients can be colonized without developing disease. The study did follow up on bloodstream infections but the overall prevalence refers to colonization, not necessarily active infection rates.

- Since this manuscript aims to address an issue that has potential of widespread attention from various field of discipline, the below listed issues would better be resolved in order to enhance the level of clarity for general readers that are outside the field of clinical microbiology. This will prevent readers from thinking that there is a logical error in their inferences.

Example: “Text 1 (Lines 85-86): States that MAs "can detect the vanA/vanB target genes from non-viable VRE isolates, potentially leading to discordant results”

>> Detecting genes from non-viable bacteria doesn't mean the patient has an active, culturable infection. The results therefore refer to a positive MA but a negative culture, which is precisely the VVE phenomenon where the gene is present but not expressed phenotypically. This is a known and accepted limitation/nuance of molecular diagnostics in clinical microbiology. From the perspective of general audience (readers), general informative basic molecular biology and microbiology information can be included.

Point by point response

We agree with the reviewer on stressing the importance of clarifying the distinction between colonization and infection (page 15, lines 380-382).

We also expanded the discussion on the potentially difficult to interpret discordant results when using traditional culture methods and MA (page 17 lines 501—514).

Reviewer 3 Report

Comments and Suggestions for Authors

This study provides important insights into VRE and VVE colonization in hospitalized patients, combining phenotypic and genomic analyses. The data are relevant and well-analyzed, though some parts, particularly the methods and figure explanations, require clarification for better readability. Here are comments from a reviewer.

Abstract

  1. Line 28 – 29: Please add the phrase “in hospitalized patients” after “to access the prevalence of VRE and VVE colonization”
  2. Line 30: Please check the correct term between “multi-locus” and “multilocus” for MLST. Based on common usage in scientific literature, multilocus is preferred. And it should be used consistently throughout the manuscript.
  3. Line 32 – 34, the statement “Of 160 rectal swabs, 54 (33.7%) tested positive for vanA and/or vanB 32 genes, and 47 (29.3%) were positive by culture: 44 isolates were identified as E. faecium and 3 33 as E. faecalis. These strains were classified as VRE in 35 cases (74.5%) and as VVE in 12 cases 34 (25.5%).” It is too dense. Please breakdown for clarity.
  4. Line 36: Please remove redundancy in “Sequence Type (ST) ST80”
  5. Please check the spelling “colonized” and “colonized” and use the term consistently throughout the manuscript.

Introduction

  1. Line 54: Please check “VREfaecium” or “VRE faecium”
  2. Line 55 – 60: The explanation van operon is too detailed and disrupts the flow of this section. The author better mention it briefly, like vanA is a part of the van operon, often carried by transposons.... The full operon explanation can be moved to the discussion section.

Results

  1. Lines 109–111: This part is clear, though how you write for result interpretation is repetitive, such as, “All isolates were found to be sensitive” or “all of them were sensitive”. Please rephrase.
  2. Figure 1A and B: Please clarify in the caption whether molecular characterization of AMR genes in VRE and VVE isolates was performed.
  3. 10. In the caption for Figure 2, please indicate the total number of isolates analyzed. Additionally, provide more explanation in the main text to help readers interpret the figure. Although some key virulence genes such as empA, empB, empC, and bepA are mentioned, they are not specifically shown in the figure—only general classifications are displayed. Please clarify where the full list of detected virulence genes can be found (e.g., in a supplementary table), and consider briefly summarizing them in the text for completeness.
  4. Add a clearer caption of figure 3

Discussion

  1. Although the study mentions vanA/B-positive PCR but culture-negative results, the diagnostic implications of this discrepancy are not fully addressed. Please discuss the risk of underestimating colonization when relying on culture alone, and the potential advantages of genomic methods in such cases.

Method

  1. Section 4.1 cgMLST analysis seems incorrect. The content is about hospital setting and PPS.
  2. Section 4.2. Rectal sampling and Phenotypic characterization of isolates paragraph contains many steps in one paragraph, which can make it hard to follow. Please breakdown this paragraph step by step or using a flow diagram figure for visual clarity.

For example: Rectal swab >>> PCR for vanA/B >>> CHROMID VRE agar >>> Columbia CAN >> MALDI-TOF >>> confirm vanA using WGS/MA >> AST

  1. The WGS and bioinformatic section is quite comprehensive, however the author can divide them into subsections for readability (e.g., assembly and annotation, pangenomes, AMR analysis, pathogen/virulence, etc).
  2. Please clarify how many genomes were used for pangenome, ANI, MST
  3. Clarify the purpose of using pyANI (e.g., species confirmation?), and specify which methodwas used (ANIb or ANIm)
  4. The statistical evaluation section is too brief. Please include details about the statistical software and specific tests used in the analysis.

Author Response

Review Report Form

Open Review

(x) I would not like to sign my review report

( ) I would like to sign my review report

Quality of English Language

( ) The English could be improved to more clearly express the research.

(x) The English is fine and does not require any improvement.

Yes         Can be improved             Must be improved          Not applicable

Does the introduction provide sufficient background and include all relevant references?

( )           (x)          ( )            ( )

Is the research design appropriate?

(x)          ( )           ( )            ( )

Are the methods adequately described?

( )           (x)          ( )            ( )

Are the results clearly presented?

( )           (x)          ( )            ( )

Are the conclusions supported by the results?

( )           (x)          ( )            ( )

Are all figures and tables clear and well-presented?

( )           (x)          ( )            ( )

Comments and Suggestions for Authors

This study provides important insights into VRE and VVE colonization in hospitalized patients, combining phenotypic and genomic analyses. The data are relevant and well-analyzed, though some parts, particularly the methods and figure explanations, require clarification for better readability. Here are comments from a reviewer.

Abstract

Line 28 – 29: Please add the phrase “in hospitalized patients” after “to access the prevalence of VRE and VVE colonization”

As suggested, we have revised the manuscript to incorporate the changes (page 1, line 29).

Line 30: Please check the correct term between “multi-locus” and “multilocus” for MLST. Based on common usage in scientific literature, multilocus is preferred. And it should be used consistently throughout the manuscript.

We have addressed this comment and made the requested changes accordingly.

Line 32 – 34, the statement “Of 160 rectal swabs, 54 (33.7%) tested positive for vanA and/or vanB 32 genes, and 47 (29.3%) were positive by culture: 44 isolates were identified as E. faecium and 3 33 as E. faecalis. These strains were classified as VRE in 35 cases (74.5%) and as VVE in 12 cases 34 (25.5%).” It is too dense. Please breakdown for clarity.

We modified the text as suggested (page 1, lines 32-36).

Line 36: Please remove redundancy in “Sequence Type (ST) ST80”

We have corrected as suggested

Please check the spelling “colonized” and “colonized” and use the term consistently throughout the manuscript.

 We have reviewed the manuscript and ensured consistent use of the term 'colonized' throughout

Introduction

Line 54: Please check “VREfaecium” or “VRE faecium”

We have consistently used the term 'VREfaecium' throughout the manuscript.

Line 55 – 60: The explanation van operon is too detailed and disrupts the flow of this section. The author better mention it briefly, like vanA is a part of the van operon, often carried by transposons.... The full operon explanation can be moved to the discussion section.

We have revised as suggested (page 2 lines 56-57 and page 13 lines 300-306).

Results

Lines 109–111: This part is clear, though how you write for result interpretation is repetitive, such as, “All isolates were found to be sensitive” or “all of them were sensitive”. Please rephrase.

We have revised where needed

Figure 1A and B: Please clarify in the caption whether molecular characterization of AMR genes in VRE and VVE isolates was performed.

We have revised the figure caption as suggested to clarify that molecular characterization of AMR genes was performed on both VRE and VVE isolates. We also modified titles and captions of all figures in response to another reviewer’s comments.

  1. In the caption for Figure 2, please indicate the total number of isolates analyzed. Additionally, provide more explanation in the main text to help readers interpret the figure. Although some key virulence genes such as empA, empB, empC, and bepA are mentioned, they are not specifically shown in the figure—only general classifications are displayed. Please clarify where the full list of detected virulence genes can be found (e.g., in a supplementary table) and consider briefly summarizing them in the text for completeness.

We thank the reviewer for this helpful suggestion. We have updated the caption of Figure 2 to indicate the total number of isolates analyzed (n = 47; 35 VRE and 12 VVE).

Further, to help readers interpret Figure 2, we expanded the caption and added supplementary Table S4, which contains the full list of detected virulence genes across all isolates.

Add a clearer caption of figure 3

We have revised the caption of Figure 3 to provide a clearer description of the data presented, as suggested.

Discussion

Although the study mentions vanA/B-positive PCR but culture-negative results, the diagnostic implications of this discrepancy are not fully addressed. Please discuss the risk of underestimating colonization when relying on culture alone, and the potential advantages of genomic methods in such cases.

We revised the text to provide more information on the challenges faced when employing molecular techniques in a diagnostic procedure (page 17 lines 501-514).

Method

Section 4.1 cgMLST analysis seems incorrect. The content is about hospital setting and PPS.

Thank you for pointing this out. We apologise for the mistake and have corrected it.

Section 4.2. Rectal sampling and Phenotypic characterization of isolates paragraph contains many steps in one paragraph, which can make it hard to follow. Please breakdown this paragraph step by step or using a flow diagram figure for visual clarity.

For example: Rectal swab >>> PCR for vanA/B >>> CHROMID VRE agar >>> Columbia CAN >> MALDI-TOF >>> confirm vanA using WGS/MA >> AST

We thank the reviewer for this helpful suggestion. To help the reader, we have included a flow diagram as a supplementary figure (Supplementary Figure S4) that visually summarizes the rectal sampling and phenotypic characterization workflow.

The WGS and bioinformatic section is quite comprehensive, however the author can divide them into subsections for readability (e.g., assembly and annotation, pangenomes, AMR analysis, pathogen/virulence, etc).

We have divided the text into subsections

Please clarify how many genomes were used for pangenome, ANI, MST

The WGS analysis was performed on all samples, we have clarified it on page 15 lines 413-414.

Clarify the purpose of using pyANI (e.g., species confirmation?), and specify which method was used (ANIb or ANIm)

We appreciate this comment, and we have clarified it on page 16 lines 438-439.

The statistical evaluation section is too brief. Please include details about the statistical software and specific tests used in the analysis.

We have expanded the statistical evaluation section by including details about the software used and the specific tests applied in our analysis. See page 17, lines 479-485.

Reviewer 4 Report

Comments and Suggestions for Authors

In the manuscript titled “Surveillance and Characterization of Vancomycin-Resistant and Vancomycin-Variable Enterococci in a Hospital Setting”, the Authors have undergone a point prevalence survey in which they have utilized rectal swabs in hospitalized patients from which they have applied a broad swathe of tests (molecular assays, culture, and subsequent whole-genome sequencing and core genome multi-locus sequence typing) to identify genetic diversity, while looking for antimicrobial resistance in the genus Enterococcus.

Relevance of the study: Antibiotic resistance in enterococci, especially Enterococcus faecium and E. faecalis, is a growing global health concern due to their role in healthcare-associated infections. These organisms can develop resistance to multiple antibiotics, including vancomycin, leaving few treatment options. Their ability to transfer resistance genes to other bacteria further amplifies the threat. A paper addressing this issue is highly relevant today, as it informs infection control strategies, guides antibiotic stewardship, and contributes to understanding resistance mechanisms. Given the rising incidence of multidrug-resistant enterococci in clinical settings, research in this area is both timely and critical for public health preparedness.

The writing is clear and to the point, showcasing precise and effective use of English. The author's message is communicated well, with no need for any changes in wording or vocabulary. The methodology is accurate, and the Conclusions sound and adequately discussed. Below are my comments and suggestions for possible improvements that could enhance the paper.

Major comments:

  • In the Abstract, the Conclusions section appears immediately after the Methods, which is somewhat unconventional. Moreover, the Conclusions begin with a summary of the study’s results. I recommend labeling this portion as a separate “Results” section. Specifically, only the sentences from lines 38 to 42 reflect the actual conclusions, while the preceding content presents the study’s findings. Please consider restructuring the Abstract to clearly distinguish the Results from the Conclusions.
  • Please mention the statistical software used for calculations.

Minor comments:

  • Please correct several typographic inaccuracies, such as “Vancomycin-Resistant” or “Point Prevalence Survey”, where the beginning letters do not need to be capitalized.
  • The Table 1 is very interesting, but contains a large amount of data which is not easily interpreted at first glance. I would suggest marking the antimicrobial resistances in another color (or shade of gray) so that they might be promptly recognized as such. This might give the reader a clearer impression on the resistance profile in the patient cohort.
  • Please check: the title and caption of Figures could be placed underneath the Figures, not above. Also, the font is small, so the schematic might benefit from enlarging the font used.

Overall, I am very satisfied with the detailed, comprehensive and thorough approach with which the Authors have approached this very important issue. The fact that it is a single-center experience on a limited number of samples should not be a problem. After addressing the Major comments, I wholeheartedly recommend it for publication in the “Antibiotics” journal.

I would like to express my gratitude to the Authors for taking the time to consider my comments and suggestions.

Author Response

Review Report Form

Open Review

(x) I would not like to sign my review report

( ) I would like to sign my review report

Quality of English Language

( ) The English could be improved to more clearly express the research.

(x) The English is fine and does not require any improvement.

Yes         Can be improved             Must be improved          Not applicable

Does the introduction provide sufficient background and include all relevant references?

(x)          ( )           ( )            ( )

Is the research design appropriate?

(x)          ( )           ( )            ( )

Are the methods adequately described?

(x)          ( )           ( )            ( )

Are the results clearly presented?

(x)          ( )           ( )            ( )

Are the conclusions supported by the results?

(x)          ( )           ( )            ( )

Are all figures and tables clear and well-presented?

( )           (x)          ( )            ( )

Comments and Suggestions for Authors

In the manuscript titled “Surveillance and Characterization of Vancomycin-Resistant and Vancomycin-Variable Enterococci in a Hospital Setting”, the Authors have undergone a point prevalence survey in which they have utilized rectal swabs in hospitalized patients from which they have applied a broad swathe of tests (molecular assays, culture, and subsequent whole-genome sequencing and core genome multi-locus sequence typing) to identify genetic diversity, while looking for antimicrobial resistance in the genus Enterococcus.

Relevance of the study: Antibiotic resistance in enterococci, especially Enterococcus faecium and E. faecalis, is a growing global health concern due to their role in healthcare-associated infections. These organisms can develop resistance to multiple antibiotics, including vancomycin, leaving few treatment options. Their ability to transfer resistance genes to other bacteria further amplifies the threat. A paper addressing this issue is highly relevant today, as it informs infection control strategies, guides antibiotic stewardship, and contributes to understanding resistance mechanisms. Given the rising incidence of multidrug-resistant enterococci in clinical settings, research in this area is both timely and critical for public health preparedness.

The writing is clear and to the point, showcasing precise and effective use of English. The author's message is communicated well, with no need for any changes in wording or vocabulary. The methodology is accurate, and the Conclusions sound and adequately discussed. Below are my comments and suggestions for possible improvements that could enhance the paper.

Major comments:

In the Abstract, the Conclusions section appears immediately after the Methods, which is somewhat unconventional. Moreover, the Conclusions begin with a summary of the study’s results. I recommend labeling this portion as a separate “Results” section. Specifically, only the sentences from lines 38 to 42 reflect the actual conclusions, while the preceding content presents the study’s findings. Please consider restructuring the Abstract to clearly distinguish the Results from the Conclusions.

We thank the reviewer for this helpful suggestion. We have restructured the Abstract to clearly separate the Results from the Conclusions. The section previously labeled as 'Conclusions' now includes a distinct 'Results' subsection summarizing the main findings, followed by a concise 'Conclusions' paragraph reflecting the key implications of the study.

Please mention the statistical software used for calculations.

We have expanded the statistical evaluation section by including details about the software used and the specific tests applied in our analysis. See pages 17, lines 479-485.

Minor comments:

Please correct several typographic inaccuracies, such as “Vancomycin-Resistant” or “Point Prevalence Survey”, where the beginning letters do not need to be capitalized.

We have carefully reviewed the manuscript and corrected all instances of unnecessary capitalization (e.g., “Vancomycin-resistant”, “point prevalence survey”, etc.)

The Table 1 is very interesting but contains a large amount of data which is not easily interpreted at first glance. I would suggest marking the antimicrobial resistances in another color (or shade of gray) so that they might be promptly recognized as such. This might give the reader a clearer impression on the resistance profile in the patient cohort.

Thank you for your helpful suggestion. We agree that enhancing the visual presentation of Table 1 would improve its readability. We have now modified the table by highlighting antimicrobial resistance results in red.

Please check: the title and caption of Figures could be placed underneath the Figures, not above. Also, the font is small, so the schematic might benefit from enlarging the font used.

Thank you for your helpful observations. We have revised the manuscript accordingly:

  • All figure titles and captions have been repositioned below the figures,
  • The font size within the figures has been increased to improve readability and ensure clarity when printed or viewed digitally.

Overall, I am very satisfied with the detailed, comprehensive and thorough approach with which the Authors have approached this very important issue. The fact that it is a single-center experience on a limited number of samples should not be a problem. After addressing the Major comments, I wholeheartedly recommend it for publication in the “Antibiotics” journal.

I would like to express my gratitude to the Authors for taking the time to consider my comments and suggestions.

Reviewer 5 Report

Comments and Suggestions for Authors

This study focuses on the prevalence of Vancomycin-Resistant Enterococci and Vancomycin-variable enterococci in a healthcare setting. While Vancomycin resistance is a significant problem, I have major concerns about the scope and experimental design of the study. Firstly, the significance of the hospital/ location for the study is missing in the introduction. This makes it difficult to assess the novelty of the study. Secondly, in line 77-79 of the introduction, it is mentioned that most studies are limited to single institutions. However, even the present study is limited to single institution. So how is it novel and significant? Moreover, how does it overcome the limitations of the previous study? Secondly, there is very little introduction/ discussion on VVE. Most importantly, in conclusion, I fail to understand lines 450-452 in conclusion. Yes, I agree this surveillance was done, but what infection control measures were taken? How would mere surveillance help in infection control?

Author Response

Review Report Form

Open Review

(x) I would not like to sign my review report

( ) I would like to sign my review report

Quality of English Language

( ) The English could be improved to more clearly express the research.

(x) The English is fine and does not require any improvement.

Yes      Can be improved          Must be improved        Not applicable

Does the introduction provide sufficient background and include all relevant references?

( )        ( )        (x)        ( )

Is the research design appropriate?

( )        ( )        (x)        ( )

Are the methods adequately described?

( )        (x)       ( )         ( )

Are the results clearly presented?

( )        ( )        (x)        ( )

Are the conclusions supported by the results?

( )        ( )        (x)        ( )

Are all figures and tables clear and well-presented?

( )        ( )        (x)        ( )

Comments and Suggestions for Authors

This study focuses on the prevalence of Vancomycin-Resistant Enterococci and Vancomycin-variable enterococci in a healthcare setting. While Vancomycin resistance is a significant problem, I have major concerns about the scope and experimental design of the study. Firstly, the significance of the hospital/ location for the study is missing in the introduction. This makes it difficult to assess the novelty of the study. Secondly, in line 77-79 of the introduction, it is mentioned that most studies are limited to single institutions. However, even the present study is limited to single institution. So how is it novel and significant? Moreover, how does it overcome the limitations of the previous study? Secondly, there is very little introduction/ discussion on VVE. Most importantly, in conclusion, I fail to understand lines 450-452 in conclusion. Yes, I agree this surveillance was done, but what infection control measures were taken? How would mere surveillance help in infection control?

We thank the reviewer for the thoughtful and constructive comments. Below we provide some detailed responses to to the main points raised:

On the significance of the hospital/location and the novelty of the study

We have clarified that the study was conducted at the National Institute for Infectious Diseases “Lazzaro Spallanzani” in Rome, Italy, which, although it is a single institution, it is one of the main national referral centres entirely dedicated to infectious diseases (page 13. Lines 290-291). For this reason, in line 291 we referred to our hospital as centre, instead of institution.

Because of this unique institutional role, our hospital receives patiens from various hospital of the Latium region, with a relatively high patient turnover. Therefore, we believe that our data provide a snapshot of the situation happening in a wider geographical area. We have clarified this concept on page 15 line 380-385.

We have also emphasized that our study is also novel in that it integrates a point prevalence surveillance with comprehensive genomic characterization (WGS, cgMLST, plasmid profiling, and virulome analysis). This level of molecular detail is rarely included in similar studies and allows us to explore intra-hospital transmission dynamics and resistance mechanisms in depth (page 17, lines 501-514).

On the limited discussion of Vancomycin-Variable Enterococci (VVE)

We have expanded the text to better define VVE, explain their clinical relevance, and highlight the diagnostic challenges they pose due to their vancomycin-susceptible phenotype despite harboring resistance genes Page 13, lines 307-315.

On the conclusion and the role of surveillance in infection control

 We have highlighted the importance of identifying VVE in colonized patients to prevent the risk of dissemination and possible reversal to a VRE phenotype of these strains (page 13, lines 309-315 and page 17, lines 507-514).

We also specified that the blood stream infections that we did not see were those caused by VRE enterococci (page 17, line 498). This information was lacking and we agree that created confusion.

Overall, the paper has been thoroughly revised and is now greatly changed. We hope that this revision addresses the reviewer’s concerns which we found useful to improve the clarity and impact of our manuscript. We are grateful for the opportunity to strengthen our work through this feedback.

Round 2

Reviewer 5 Report

Comments and Suggestions for Authors

The authors have addressed my concerns.